# Adipose Tissue, Non-Communicable Diseases, and Physical Exercise: An Imperfect Triangle

**DOI:** 10.3390/ijms242417168

**Published:** 2023-12-06

**Authors:** Francisco A. Monsalve, Fernando Delgado-López, Barbra Fernández-Tapia, Daniel R. González

**Affiliations:** 1Department of Basic Biomedical Science, Faculty of Health Sciences, Universidad de Talca, Talca 3465548, Chile; dagonzalez@utalca.cl; 2Laboratories of Biomedical Research, Department of Preclinical Sciences, Faculty of Medicine, Universidad Católica del Maule, Talca 3466706, Chile; fdelgado@ucm.cl; 3Nursing School, Faculty of Health Sciences, Universidad de Talca, Talca 3465548, Chile; bfernandez@utalca.cl

**Keywords:** adipose tissue, adipocytokines, overweight, obesity, molecular mechanism, treatments, myokines

## Abstract

The study of adipose tissue has received considerable attention due to its importance not just in maintaining body energy homeostasis but also in playing a role in a number of other physiological processes. Beyond storing energy, adipose tissue is important in endocrine, immunological, and neuromodulatory functions, secreting hormones that participate in the regulation of energy homeostasis. An imbalance of these functions will generate structural and functional changes in the adipose tissue, favoring the secretion of deleterious adipocytokines that induce a pro-inflammatory state, allowing the development of metabolic and cardiovascular diseases and even some types of cancer. A common theme worldwide has been the development of professional guidelines for the control and treatment of obesity, with emphasis on hypocaloric diets and exercise. The aim of this review is to examine the pathophysiological mechanisms of obesity, considering the relationship among adipose tissue and two aspects that contribute positively or negatively to keeping a healthy body homeostasis, namely, exercise and noninfectious diseases. We conclude that the relationship of these aspects does not have homogeneous effects among individuals. Nevertheless, it is possible to establish some common mechanisms, like a decrease in pro-inflammatory markers in the case of exercise, and an increase in chronic inflammation in non-communicable diseases. An accurate diagnosis might consider the particular variables of a patient, namely their molecular profile and how it affects its metabolism, routines, and lifestyle; their underling health conditions; and probably even the constitution of their microbiome. We foresee that the development and accessibility of omics approaches and precision medicine will greatly improve the diagnosis, treatment, and successful outcomes for obese patients.

## 1. Adipose Tissue

Beyond storing energy, adipose tissue is important in several processes, such as the modulation of energy homeostasis, metabolism, and the regulation of the immune system [1]. Adipose tissue is considered an endocrine organ since it produces and secrets molecules that can exert their action in surrounding or distant tissues [2]. Adipose tissue is composed of adipocytes, pre-adipocytes, endothelial cells, fibroblasts, and some immune cells such as macrophages, dendritic cells, and T cells that contribute to the release of metabolites, lipids, cytokines, and adipocytokines [3]. Hormones and adipocytokines produced by adipocytes affect the central nervous system, skeletal muscle, liver, bone, and other tissues, which have been studied extensively in the last two decades, establishing that these factors play a preponderant role in the homeostasis of body glucose, through endocrine, autocrine, and paracrine mechanisms. Adipocytokines are essential for the balance between appetite and satiety, body fat reserve and energy expenditure, glucose tolerance, insulin release and sensitivity, cell growth, inflammation, angiogenesis, and reproduction [4].

Under physiological conditions, adipose tissue plays a central role in keeping the homeostasis of the entire body serving as the main storage for excess energy; namely energy that is used during fasting, thus preserving proteins, regulating metabolism, satiety, reproduction, and enhancing the immune response in pathogenic invasion [1].

Adipocytes that have a large drop of lipids, called unilocular adipose cells, form part of the white adipose tissue and cells with multiple small drops of lipids, called multilocular adipose cells, constitute the brown adipose tissue [5]. White adipose tissue is the most abundant and is distributed throughout the body, mainly as perivascular and visceral fat [3]. It produces and secretes adipocytokines, glucocorticoids, and sex hormones [6]. Brown adipose tissue is considered thermogenic and its color reflects its numerous mitochondria [7]. UCP1 decoupling protein is responsible for modifying oxidative phosphorylation in mitochondria, causing a decrease in ATP production and increasing heat production by this tissue [8]. Brown adipose tissue has a regulatory function in body temperature via adaptive thermogenesis, regulating the concentration of circulating triglycerides, storing glucose, and secreting prostaglandins, nitric oxide, adipsin, and other adipocytokines [9].

The beige adipose tissue has an intermediate diameter between white and brown adipose tissue [7]. Originally, it was observed mainly in response to cold [10]. However, factors such as diet, physical, pre and probiotic activity, and drugs, among others, are able to induce the transdifferentiation (to beige or brown) of white adipose tissue. This tissue has the function of storage or energy expenditure, according to physiological needs [8].

The excessive adiposity that occurs in obesity (excess of white adipose tissue) disrupts the metabolic balance of the adipose tissue, producing a negative impact on the homeostasis of the human body. Therefore, obesity is the causality of a group of chronic and complex diseases such as cardiovascular diseases, metabolic syndrome, type 2 diabetes [11], and even some types of cancer [12].

## 2. Adipose Tissue as an Endocrine Organ

Adipose tissue, besides being an energy reservoir, is an important endocrine organ as it produces several hormones that participate in the regulation of homeostasis. A positive imbalance of adipose tissue (accumulation of fatty acids over a BMI 24.9 kg/m^2^) generates structural and functional changes in this tissue, favoring the secretion of deleterious adipocytokines related to insulin signaling and those that benefit a pro-inflammatory state, which could promote the development of metabolic and cardiovascular diseases (see Figure 1) [13]. One of the first of these hormones discovered was leptin [14], which suppresses food intake by inducing satiety, along with increasing energy expenditure. Leptin levels are positively correlated with the amount of adipose tissue and it is secreted mainly by visceral white adipose tissue [15,16]. Adiponectin is a hormone secreted by subcutaneous white adipose tissue which has anti-inflammatory and insulin-sensitizing functions [17]. In overweight or obese people with insulin resistance, plasma adiponectin levels are low [18]. Resistin is another hormone secreted by adipose tissue and has a close relationship with obesity and diabetes, notably contributing to insulin resistance and vascular inflammation [19]. Fibroblast growth factor 21 (FGF21) is a protein produced by adipose and other tissues, with thermogenic effects that promote the transdifferentiation from beige to brown adipose tissue. Like adiponectin, it has insulin-sensitizing effects and in overweight or obese patients, plasmatic concentrations are elevated [13]. Vaspin (also serpin A12) is an inhibitor of serine protease, which acts as an insulin-sensitizing adipocytokine [20,21], that is increased in obese patients, promoting insulin resistance and decreasing glucose tolerance [22]. Visfatin is another hormone involved in glucose homeostasis and, like acylation-stimulating protein (ASP), is preferentially involved in fat storage [23].

Adipose tissue secretes inflammatory cytokines such as interleukin-6 (IL-6), IL-8, interferon-γ (INF-γ), and plasminogen activator inhibitor-1 (PAI-1) [24]. Other molecules secreted by adipose tissue include retinol-binding protein 4 (RBP4), omentin, angiotensinogen, macrophage migration inhibitory factor (MIF), lipoprotein lipase (LPL), prostaglandins, estrogens, and glucocorticoids [14]. All these molecules influence homeostatic processes, affecting health positively or negatively and resulting in the development of several diseases, such as type 2 diabetes mellitus (T2DM), metabolic syndrome, and several types of cancer (breast, cervical, endometrial, renal, and gastrointestinal). Even more, adipose tissue dysfunction can lead to psychiatric disorders, such as depression, dementia, insomnia, and many other [1].

## 3. Definition and Incidence of Overweight and Obesity

The Body Mass Index [BMI (kg/m^2^, weight of the person divided by the square of their height)] is used to define and diagnose obesity according to the clinical guidelines of the World Health Organization (WHO) [25]. In adults, the WHO defines overweight as a BMI of 25.0 to 29.9 kg/m^2^ and obese as a BMI > 30.0 kg/m^2^. In addition, obesity is classified into three levels of severity: class I (BMI 30.0–34.9 kg/m^2^), class II (BMI 35.0–39.9 kg/m^2^), and class III (BMI > 40.0 kg/m^2^) [26]. For every 5-unit increase in BMI above 25.0 kg/m^2^, overall mortality increases by 29%, vascular mortality by 41%, and diabetes-related mortality by 210% [27].

Obesity is often stigmatized and associated with a false perception that it is primarily caused by a lack of willpower, leading to inappropriate dietary choices and physical inactivity. However, there is abundant literature-based evidence presenting obesity as a complicated chronic medical condition caused by multiple genetic, environmental, metabolic, and behavioral factors [28].

Obesity increases the probability of developing several other diseases and pathological conditions that are linked to increased mortality, such as type 2 diabetes mellitus (T2DM), cardiovascular disease (CVD), metabolic syndrome (MetS), chronic kidney disease (CKD), hyperlipidemia, hypertension, non-alcoholic fatty liver disease (NAFLD), certain types of cancers, obstructive sleep apnea, osteoarthritis, depression, and neurodegenerative disorders [29].

The pathogenesis of obesity is complex, with environmental, socio-cultural, physiological, medical, genetic, epigenetic, and numerous other factors contributing to the cause and persistence of this condition [30].

## 4. Causes or Mechanisms of Obesity

### 4.1. Genetic Factors

Data show that around 40 to 70% of the variations in obesity in humans are the result of genetic factors [31]. While environmental changes have increased the rate of obesity, genetic factors play a key role in the development of this condition, with nearly 100 genes related to obesity and fat distribution [28,32]. It is clear that in the same environment, some people become obese and others do not. Many factors are involved in this differential response and some of them will be mentioned below.

Genetic causes of obesity can be classified as (A) monogenic causes that result in a single mutation located mainly in the leptin–melanocortin pathway. Many of the genes, such as AgRP (Agouti-related peptide), PYY (peptide tyrosine tyrosine, and orexigenic), or MC4R (melanocortin-4 receptor), were identified as causing monogenic obesity and deregulating appetite and body weight control systems, where hormonal signaling (ghrelin, leptin, and insulin) is sensed by receptors located in the hypothalamus (arcuate nucleus) [33]. (B) Syndromic obesity, as the result of severe obesity due to neurodevelopmental abnormalities and other organ/system malformations. This can be caused by mutations in a single gene or a chromosomal region that spans multiple genes [34]. (C) Polygenic obesity, caused by the cumulative contribution of several genetic alterations. The presence of these types of alterations causes an increase in caloric intake, an increase in appetite, a reduced control of satiety, and a higher tendency to store body fat and to a sedentary lifestyle [35].

There are also several genetic, neuroendocrine, and chromosomal syndromes that cause obesity, such as Prader-Willi syndrome (PWS), which is a neurodevelopmental disorder involving hypothalamic dysfunction and leading to impaired secretion of several hormones [36] and polycystic ovary syndrome, an endocrine disorder that results in increased body fat mass [37] associated with deletions such as 16p11.2, 2q37 (brachydactylic mental retardation syndrome), 1p36 (monosomy 1p36 syndrome), 9q34 (Kleefstra syndrome), 6q16 (PWS-like syndrome), 17p11.2 (Smith Magenis syndrome), and 11p13 (WAGR syndrome) [38]. All of these conditions show an energy imbalance between calories intake and expenditure as the main cause of obesity [39].

### 4.2. Fat Cells

The excess of calories from food intake results in an accumulation of fat in adipocytes [28]. This enlargement and/or increase in the number of fat cells to adapt to increased fat storage establish the initial pathological lesion in obesity. The accumulation of ectopic fat, such as visceral, cardiac, and muscle fat, is associated with several factors when adipocytes have already reached their maximal storage capacity [40]. Nevertheless, as the group of Scherer showed, a complete abolition of adipose tissue involved in inflammation was associated with several adverse metabolic readouts [41]. In other words, inflammation in a healthy dose is important during adipose expansion. The increase in the size of the adipocyte eventually generates an inflammatory microenvironment due to an alteration of the homeostasis between the adipocyte and the surrounding cells, mainly resident macrophages [4]. The alteration of a healthy expansion in adipocyte size produces an increase in the secretion of several inflammatory adipocytokines (some seen in the Section 2), such as leptin, IL-6, TNF-α, angiotensinogen, adipsin, free fatty acids, and lactate, while the levels of anti-inflammatory molecules secreted, such as adiponectin, decreases [42].

### 4.3. Dysregulation of Energy Balance

Genes and the environment interact in a complex manner in physiological processes that regulate energy balance and body weight [43]. Two groups of neurons located in the arcuate nucleus of the hypothalamus are inhibited or stimulated by ghrelin and leptin, they are thus hormones that control energy balance by regulating food intake and energy expenditure, namely AgRP and POMC neurons. Brain regions external to the hypothalamus also contribute to the regulation of energy balance through sensory signals, cognitive processes, memory, and attention [44].

Reducing food intake or increasing physical activity generates a negative energy balance, activating adaptive compensatory mechanisms that preserve vital functions [45]. Conversely, at rest, there is a relative reduction in energy expenditure, seeking food and metabolic processes that depend on the magnitude and duration of caloric restriction [46]. An increase in the stimulation of the orexigenic center could explain a subtle and often inappropriate increase in appetite and food intake, limiting weight loss associated with interventions such as physical exercise programs. It is important to always consider obesity as a chronic disease, that requires long-term monitoring and weight control since there is a high relapse rate in those people who have managed to lose weight [30].

### 4.4. Metabolic and Physiological Effects

As mentioned, adipocytes synthesize signaling molecules (adipocytokines) and hormones and their secretion and effects are influenced by the distribution and amount of adipose tissue in the body [47]. The excessive secretion of proinflammatory adipocytokines by adipocytes and macrophages within adipose tissue leads to low-grade systemic inflammation in people with obesity [30].

Triglyceride breakdown in adipocytes leads to the release of free fatty acids which are then transported in the plasma to sites where they can be metabolized. In overweight people and to a greater extent in those with obesity, free fatty acid levels are often elevated, reflecting the increased mass of adipose tissue [47].

Lipids are not only stored in adipose tissue; they are also stored in other cell types in organelles called liposomes, located near mitochondria [48]. For example, liposomes in hepatocytes increase in size, forming large vacuoles, which are observed in a series of pathological states such as non-alcoholic fatty liver disease, steatohepatitis, and cirrhosis [49], generating cell dysfunction and apoptosis [30].

Elevated levels of free fatty acids, proinflammatory cytokines, and intermediary lipids, such as ceramides, in non-adipose tissues, contribute to impaired insulin signaling and a state of insulin resistance [50]. All these metabolic and anatomical findings are some of the pathophysiological mechanisms caused by dyslipidemia in obesity, type 2 diabetes, obesity-related liver disease, and osteoarthritis; they are also implicated in the development of some cancers, likely owing to the association with elevated levels of tumor-promoting molecules [12].

There is cumulative evidence showing a complex interplay between obesity and both the central nervous system and the peripheral nervous system. These associations are quite complex because they not only involve the so-called organokines (adipokines, myokines, and hepatokines) acting on the nervous system but they also involve hormones and factors secreted by the nervous tissue acting on other organs. Furthermore, there is also evidence that associates inflammation-related obesity to a leaky gut [51,52,53], with alterations in the gut microbiota, through which intact Gram (−) bacteria or broken-down products of its wall pass through the intestinal epithelium and reach the bloodstream [54]. The wall of Gram (−) bacteria is rich in lipopolysaccharides, also called endotoxin, which is a Pathogen Associated Molecular Pattern (PAMP) recognized by the Toll-like receptor 4, a member of the Toll family of receptors, involved in triggering pro-inflammatory signaling through NF-kappa B transcription factor [55] that activates the expression of several genes associated with inflammation, like cytokines and chemokines. Of note, humans are significantly more sensitive to LPS than other species [56]. Finally, sympathetic nervous system hyperactivity in some overweight or obese individuals might produce multiple pathophysiological processes (see Figure 2) such as arterial hypertension, heart disease, a heart attack, and chronic kidney disease, all associated with insulin resistance, dyslipidemia, and type 2 diabetes [57].

## 5. Complications and Comorbidities Associated with Obesity

### 5.1. Insulin Resistance (IR)

Obesity is associated with increased mortality. For every 5 kg/m^2^ increase in BMI above 25 kg/m^2^, the overall mortality increases by approximately 30%, vascular mortality by 40%, and diabetic, renal, and hepatic mortality by 60% to 120%. With a BMI of 30 to 35 kg/m^2^, the median survival is reduced by 2 to 4 years, and a BMI of 40 to 45 kg/m^2^ by 8 to 10 years [58]. The main cause of death in obesity includes cardiac ischemia and complications associated with diabetes [59], including components of insulin resistance (IR) as well as metabolic syndrome (hypertension, hyperglycemia, and dyslipidemia, among others) [60].

Obesity is associated with an increased risk of IR. The HOMA–IR (Homeostatic Model Assessment–IR) relationship is strongly correlated with visceral fat mass, total fat, and waist circumference [61]. Adipose tissue controls metabolism by regulating the levels of non-esterified fatty acids (NEFAs), glycerol, proinflammatory cytokines, immune system cells (lymphocytes and macrophages), and several hormones [62]. Many of these molecules are increased in obesity, affecting insulin sensitivity through various mechanisms. First, the increased transport of NEFAs and consequently high intracellular levels compete with glucose for the oxidation of substrates, resulting in the inhibition of important enzymes involved in the glycolytic pathway [63]. In addition, metabolites from NEFAs (ceramides, diacylglycerol, and acyl-coenzyme A, among others) are increased, affecting the insulin receptor signaling pathway by phosphorylation of the insulin receptor substrate type 1 and 2 (IRS-1 and -2), which reduces the activity of phosphatidyl-3-kinase (PI3K) [64]. Second, an increase in proinflammatory cytokines, as well as immune cells, overstimulates inflammatory processes, causing tissue dysfunction, hypoxia, and injury. Obesity generates a chronic low-grade inflammation, increasing the secretion of TNF-α, IL-6, and MCP-1, mainly by visceral adipose tissue [65]. Proinflammatory signaling pathways involve JNK activation and IKK inhibition, again leading to phosphorylation of IRS-1 and IRS-2 as well as increased transcription of proinflammatory genes [66]. And third, increased levels of proteins such as RBP-4 (retinol-binding protein-4) and leptin and reduced levels of adiponectin affect insulin sensitivity by altering the PI3K signaling pathway in muscle, inducing the expression of gluconeogenic enzymes and the oxidation of fatty acids in the liver, giving liver and muscle IR as a final result, causing abnormal glucose production from the liver [67] and reduced glucose uptake by skeletal muscle [68]. See Figure 3.

### 5.2. Type 2 Diabetes

Obesity is strongly associated with the development of type 2 diabetes (T2DM): the insulin-resistance state demands an increase in insulin delivery to the liver, muscle, and adipose tissue to maintain euglycemia [69]. β-pancreatic cells are able to increase their functionality and mass to meet the increased demand for a limited time, then turning to dysfunction, and finally, loss of part of these cells [60,70]. Muscular IR affects the metabolism of the whole organism since skeletal muscle is the major site of postprandial glucose uptake (80%) and thus, insulin resistance at this site is a substantial contributor to the development of T2DM [71], promoting hepatic steatosis and an increase in adipose tissue [62]. In addition, in the liver, insulin finely regulates postprandial glucose levels by suppressing hepatic glucose production and stimulating glucose storage as glycogen [72,73]. In patients with T2DM, insulin cannot regulate glycogen synthesis or glucose production, leading to increased hepatic gluconeogenesis as the main cause of hyperglycemia in T2DM [67]. The combination of hyperglycemia and dyslipidemia (glucolipotoxicity) accelerates β-cell death, reducing insulin secretion, and further aggravating hyperglycemia [74]. As a consequence, the relative risk of incidence of diabetes is approximately 1.87 per standard deviation of BMI or the waist/hip ratio [75].

### 5.3. Dyslipidaemia

Obesity is associated with dyslipidemia, which is characterized by increased plasma levels of total cholesterol, triglycerides, VLDL-apoB, postprandial hyperlipidemia, lower HDL cholesterol (HDL-c) levels, and a predominance of small dense LDL cholesterol particles [60,76]. The accumulation of lipolytic active visceral fat in combination with IR leads to a marked increase in the delivery of fatty acids to the portal vein and subsequently, an increase in the hepatic synthesis of triglycerides, with a higher risk of coronary disease in those patients with hypertriglyceridemia [77]. IR and low HDL-c levels present a common mediator, TNF. TNF is involved in insulin resistance in obese patients and is recognized for lowering HDL-c levels [78]. Several enzymes involved in HDL-c metabolism are altered in IR [79], which further increases the atherogenic risk.

### 5.4. Hypertension

Several mechanisms have been implicated in obesity-associated hypertension. First, obesity is characterized by hemodynamic changes due to volume overload, causing increased cardiac output, peripheral vascular resistance, and increased arterial blood pressure [80,81,82,83]. Second, high salt intake due to increased food consumption alters sodium homeostasis [84], promoting hypertension [85]. Higher sodium reabsorption combined with higher renal blood flow plus hyperfiltration leads to changes in renal structure and dysfunction, contributing to further elevation of blood pressure [86]. In addition, hormonal changes that are observed in obesity such as hyperaldosteronism, hyperinsulinemia, and hyperleptinemia result in activation of the renin–angiotensin–aldosterone system (RAAs) [87], the sympathetic autonomic nervous system, and decreased parasympathetic activity [88,89]. And third, the increases in the production of reactive oxygen species [90] in combination with endothelial dysfunction with vascular stiffness, reducing the bioavailability of nitric oxide [91], leads to decreased acetylcholine-dependent endothelial relaxation [92]. All these hormonal and vascular changes, plus the low-grade chronic inflammation that patients with obesity present with, leads them to suffer from hypertension.

The high risk of cardiovascular events such as heart attack, heart failure, or cardiac arrest, would be partly explained by IR, dyslipidemia, hypertension, hyperglycemia, and diabetes that obese patients can present with, mediating 44% of the risk of coronary disease and 69% of the risk of cardiac arrest [93].

### 5.5. Other Complications and Comorbidities

Other complications and comorbidities associated with obesity are (a) Polycystic Ovarian Syndrome. The worldwide increase in the prevalence of obesity has led to an increase in comorbidities such as polycystic ovary syndrome (PCOS). In a genetic susceptibility setting, PCOS often manifests after weight gain, commonly in adolescence. PCOS is a common endocrinopathy that affects between 6 and 10% of women of childbearing age and presents characteristics such as hyperandrogenism and metabolic and reproductive dysfunction, as well as IR, independent of obesity, although it is amplified when it is present [94]. Although the mechanism of IR in PCOS is not fully elucidated, the reported defects would be within the insulin receptor signaling pathway and low-grade inflammation that occurs in PCOS [95,96]. (b) Obstructive Sleep Apnea. Obstructive sleep apnea (OSA) occurs in a high prevalence in patients with obesity and coincides with several comorbidities such as hypertension, type 2 diabetes, dyslipidemia, non-alcoholic fatty liver disease, heart failure, and atrial fibrillation [97]. It has been estimated that 58% of moderate to severe OSA is due to obesity [98] and OSA is an independent risk factor for stroke [99]. However, one should be aware of other risk factors, such as advanced age, male gender, peri- or postmenopausal states in the female gender, and craniofacial abnormalities [100]. (c) Cancer. Obesity is strongly related to increased susceptibility to various diseases, including different types of cancers, such as thyroid, uterine, liver, pancreatic, colorectal, breast, esophageal, and kidney cancers [12,60]. It is estimated that one in five cancers is related to obesity [101]. Several mechanisms linking cancer to obesity have been proposed, such as IR, high levels of IGF-1, chronic low-grade inflammation in obese patients, deregulation of factors/hormones secreted by adipose tissue, and alterations in sex hormones [102], as well as changes in the population of immune cells [12]. See Figure 4.

## 6. Management of the Overweight-Obesity Patient

Weight loss should be recommended for all obese patients and also for overweight individuals with comorbidities such as insulin resistance, diabetes, hypertension, and dyslipidemia [103]. For many patients who need to lose weight for medical reasons, the initial goal is to lose 5–10% of their body weight in the first six months [104]. A common theme worldwide has been the development of professional guidelines. In the case of Chile, the Chilean Society of Metabolic and Bariatric Surgery led the process of adapting the adult obesity clinical practice guideline [105], which establishes changes in the approach to managing obesity as a chronic disease, with emphasis on multifactorial lifestyle interventions that include dietary changes, increase in physical activity, and behavior modifications. Pharmacotherapy, medical devices, and bariatric surgery are other options for patients who need additional interventions [106].

### 6.1. Lifestyle Modification

To successfully achieve a significant 5–10% in weight loss requires comprehensive and intensive patient intervention within the first six months [104]. Effective weight loss interventions include dietary modifications through the prescription of low-calorie diets, increased physical activity or exercise, and behavioral strategies to encourage adherence to the dietary and physical activity recommendations [107]. Common strategies include self-monitoring of diet and physical activity, daily or regular weighing, goal setting, and stimulus control, such as limiting eating places [108]. Significant weight loss in a short time can be achieved by the controlled consumption of small portions of food [109]. Long-term weight control can be achieved via high levels of physical activity and continuous doctor–patient contact. In many cases, lifestyle modification results in a dramatic loss of body weight, leading to a significant reduction in cardiovascular risk [110].

### 6.2. Pharmacotherapy: Who Are Candidates to Receive Anti-Obesity Drugs?

Drugs approved for the management or control of body weight should be considered for patients with a BMI ≥ 30 kg/m^2^ and those with a BMI of at least 27 kg/m^2^ in the presence of weight-related comorbidities [104]. Pharmacotherapy may be considered for patients with excess body weight who (i) achieve a modest benefit with lifestyle intervention and require further reduction in body weight; (ii) who have lost some weight with the lifestyle intervention but are having difficulty maintaining that loss; (iii) you have made numerous unsuccessful attempts to lose weight through diet and exercise; and (iv) are unable to comply with lifestyle change recommendations due to serious chronic medical conditions (see Figure 5) [103].

### 6.3. Orlistat

Orlistat is a pancreatic lipase inhibitor that reduces intestinal fat absorption [103], resulting in approximately 30% of ingested triglycerides being eliminated in the stools. To date, it is the only drug available for obesity that does not target satiety or appetite mechanisms [111]. Orlistat 120 mg three times a day for one year achieves a reduction in body weight of approximately 3% [112] and is approved for patients with BMI ≥ 27 kg/m^2^ in the presence of comorbidities such as hypertension, type 2 diabetes, and dyslipidemia, among others [111]. Orlistat in low doses (60 mg three times a day) managed to reduce body weight between 1.6% to 2.4% in six months of treatment [103].

Treatment with orlistat generates gastrointestinal adverse effects, such as oily staining and oils in stools, farting with discharge, fecal urgency, and increased defecation [113]. These side effects may cause patients to discontinue the treatment.

### 6.4. Liraglutide

Liraglutide is a subcutaneous injectable Glucagon-Like Peptide-1 (GLP-1) agonist approved in 2010 for the treatment of T2DM at a dose of 1.8 mg per day. The data show that liraglutide acts on POMC/CART (pro-opiomelanocortin/the cocaine-and amphetamine-regulated transcript) neurons, decreasing appetite and increasing levels of satiety in addition to generating a transient effect of slowing gastric emptying [114,115]. Another effect of liraglutide is to increase insulin release and suppress glucagon secretion in fasting and postprandial glycaemia in subjects with T2DM [116]. Nausea, vomiting, heartburn, constipation, and diarrhea are the most common adverse effects of liraglutide, especially nausea, which has an incidence of at least 40%, which is due to a transient decrease in gastric emptying [103,117].

### 6.5. Phentermine

Phentermine, chemically similar to amphetamine, is a sympathomimetic amine with an anorexigenic effect, which increases the secretion of norepinephrine and dopamine in the central nervous system. The action of phentermine is due to the inhibition of neuropeptide Y/agouti-related peptide (NPY/AGRP) secreting neurons and an increase in energy expenditure [118].

Phentermine is approved for use for three months. Formulations of 18.75 and 37.5 mg are found, although a 30 mg prolonged-release formulation was recently incorporated [111]. Headache, insomnia, increased blood pressure, tachycardia and palpitations, rhabdomyolysis, and intracranial bleeding/stroke are the most common adverse effects of phentermine use [119,120].

### 6.6. Naltrexone/Bupropion

Bupropion, an approved smoking-cessation antidepressant agent, has been shown to promote significant weight loss in obese patients, inducing satiety by enhancing the production and secretion of α-MSH (α-melanocyte-stimulating hormone) and β-endorphins from pro-opiomelanocortin cells in the arcuate nucleus of the hypothalamus [121]. Naltrexone, an antagonist of µ-opioid receptor, used for the treatment of alcohol and opiate dependence, can decrease food cravings and intake, causing weight loss in subjects receiving this treatment [122]. The most common adverse effects of the naltrexone/bupropion combination are nausea, constipation, headache, vomiting, insomnia, dry mouth, dizziness, and diarrhea [105].

### 6.7. Phentermine/Topiramate

As described above, phentermine has sympathomimetic actions, a noradrenergic agonist, which increases the secretion of noradrenaline, dopamine, and serotonin [123]. Topiramate is currently used in the treatment of epilepsy and migraines since it is an α-amino-3-hydroxy-5-methyl-4-isoxazole propionic acid/kainate (AMPA/KA) glutamate receptor antagonist. Topiramate has a GABAergic mechanism of action but it is also an anorexigenic and is currently used for the treatment of obesity [124].

The most common adverse effects are paresthesia, dry mouth, constipation, insomnia, dizziness, and dysgeusia [125].

### 6.8. Diethylpropion

Diethylpropion is an anorexigenic agent with a mechanism of action similar to the antidepressant agent bupropion. The drugs used for the treatment of obesity are commonly called anorexigenics and their primary action is the suppression of appetite, either at the central or metabolic level. Diethylpropion acts at the level of the central nervous and cardiovascular systems, raising blood pressure by inhibiting the reuptake of dopamine and norepinephrine [126].The most common side effects are dry mouth, insomnia, and stimulation of the nervous system [126]. Therefore, extreme precautions should be taken when using this drug since it may generate dependence [127].

## 7. Physical Exercise as Treatment

The benefits of physical exercise have been shown to prevent all causes of mortality, including cardiovascular diseases, metabolic diseases, and cancer [128]. Exercise reduces the risk of death by preventing the development of metabolic pathologies and protecting against chronic diseases. Organ–organ interaction involves muscle contraction at the molecular level as an emerging exercise-related field (see Figure 6) [129]. In addition, adipocytokines mediate the relationship between adipose tissue and other tissues, such as the brain, as well as metabolic functions during tissue activation [130]. The pro-inflammatory role of adipocytokines has been well identified (see above) and their secretion is increased in obesity, causing metabolic and cardiovascular diseases, among others [128].

The benefits induced by physical exercise are well known to prevent the harmful effects of proinflammatory adipocytokines through proteins secreted by the muscle (myocytokines or myokines) [129]. In recent years, several studies have shown that acute aerobic exercise is an important mediator of the systemic anti-inflammatory response [131]. Muscle contraction induced by physical exercise, regardless of the intensity, volume, or density, produces an increase in gene expression and secretion of IL-6 into plasma [132]. After intense physical exercise, IL-1 receptor antagonist (IL1-ra) and soluble tumor necrosis factor I and II receptors (TNF I and II) are increased in plasma. This set of changes is the so-called “anti-inflammatory effect” [133].

Given this condition, the proposed hypothesis is that regular physical exercise (chronic effect) exerts an anti-inflammatory effect leading to protection against chronic inflammation produced by proinflammatory cytokines and C-reactive protein [132]. However, most adults who have chronic non-communicable diseases do not reach exercise levels according to guidelines [105], in addition to a reduction in physical activity levels and an increase in sedentary behavior [134]; therefore, overweight or obese adults should consider increasing their physical activity as part of comprehensive obesity treatment. Physical exercise offers a wide range of health benefits and with just 150 min of physical activity a week (walking, cycling, or resistance exercises), the expected benefits can be obtained.

## 8. Hypocaloric Diets as the Most Effective Approach

There is no question of the overall benefits of exercise in general; as for aerobic exercise in particular, its association with improving most functions and biochemical parameters associated with a healthy physiology, overall wellbeing, and extending people’s lifespan is clear. However, exercise as the main approach to lose weight is not very efficient, since to burn 1 kg of fat the use of 7700 calories is required; for most untrained people, burning 350 calories daily just by doing exercise is not an easy endeavor, both due to the exertion involved and the time required to perform a given routine on a daily basis [135]. That is explained by the fact that most of the daily energy requirements are used to maintain the body temperature and much less is used by our daily physical activity [136].

Hence, there is abundant evidence showing that a hypocaloric diet is, in most cases, the best standalone approximation to reduce obesity, albeit improved if accompanied by some exercise. It is interesting to note that mice on low-calorie diets live up to twice as long as their normal diet counterparts, an observation whose mechanisms are not well understood [137,138] and for which there is no evidence in humans. It is important to note that the range of hypocaloric programs is wide in several ways, length, nutrients, schedule, and perhaps most important, the maximum number of calories allowed daily [139]. The choices will also depend on the patient’s particulars, like BMI, age, comorbidities, and biochemical blood parameters. Some, particularly those under 1000 calories per day, will require close medical supervision and some will even be followed as in-patients at a hospital or related institution [140].

## 9. Conclusions

Obesity is perhaps the single most extensive, expensive, and penetrant condition or disease of the present time, caused by a multitude of factors, from genetics to the environment. Initially, the pathophysiology of obesity was discussed and understood in the context of excessive caloric intake and its accumulation within the adipose tissue. Currently, it has become evident that the interplay between all systems needs to be considered to understand its pathophysiology and to approach effective treatments. A paradigm that considers the contributions of secreted molecules, such as proteins, hormones, nucleic acids, and metabolites by different tissues, and affects the functions and communication of adipose cells with the rest of the systems will be central to comprehend their particular role in sustaining healthy metabolic homeostasis or not.

A more detailed understanding of the several dimensions of obesity, including predisposition to regain weight and individual differences in the pathogenesis and response to treatments, is necessary for designing effective interventions, especially to address the additional complications associated with obesity such as diabetes, hypertension, and kidney diseases, among others.

Small changes in physical activity produce significant benefits in patients. It is well known that physical activity or regular exercise prevents the development of chronic diseases, with an impact on low-grade chronic inflammation. Muscle contractions associated with physical activity or exercise promote the secretion of myokines, which could be potential candidates to provide beneficial effects by stimulating metabolic pathways, improving glucose uptake and fatty acid oxidation as well as muscular growth and regeneration. Additionally, the synergy between exercise and mental and physical health must not be underscored.

Despite intense investigation in the area of obesity, the precise pathophysiological mechanisms and the individual and specific differences and how they could be tailored in terms of gender, ethnicity, or age groups still remains unclear. The advent and eventual synergy of all the omics, artificial intelligence, and other technological innovations will aid in this quest, most likely through finding complex mechanistic interactions that could guide precise therapy. There is not a clear-cut correlation as to how chronic inflammation is affected by exercise or how the inflammation associated with underlying non-communicable diseases interplay in obesity. Therapeutic strategies aimed at preserving or restoring adipose tissue functionality hold numerous unknown opportunities to improve human wellbeing. Nonetheless, significant efforts should be globally agreed in terms of education and lifestyle changes toward having more harmonious and sustainable lifestyles, where prevention efforts should diminish the need for reactive interventions.

## Figures and Tables

**Figure 1 ijms-24-17168-f001:**
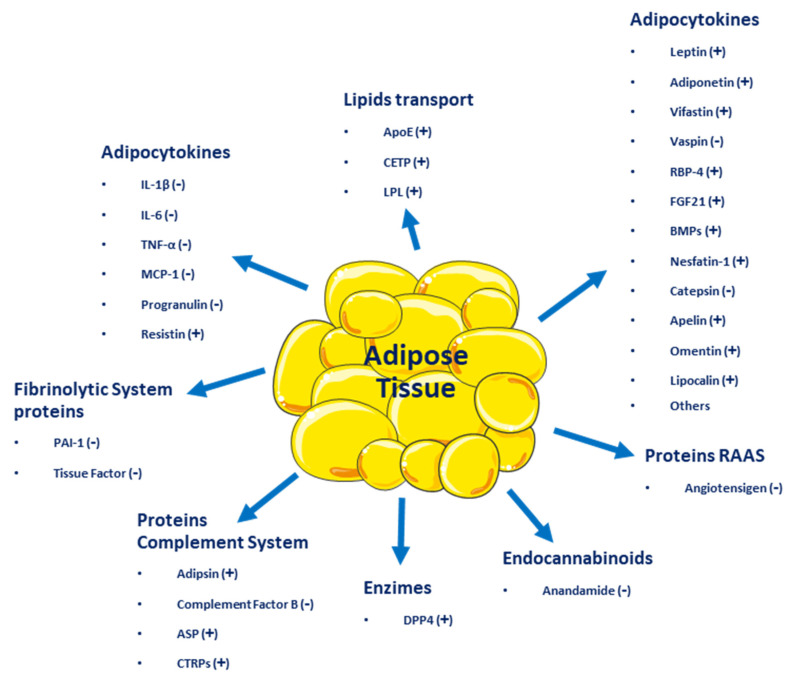
Adipocytokines and other molecules are secreted by adipose tissue. (+) Beneficial effect on energy homeostasis; (−) Negative effect on energy homeostasis.

**Figure 2 ijms-24-17168-f002:**
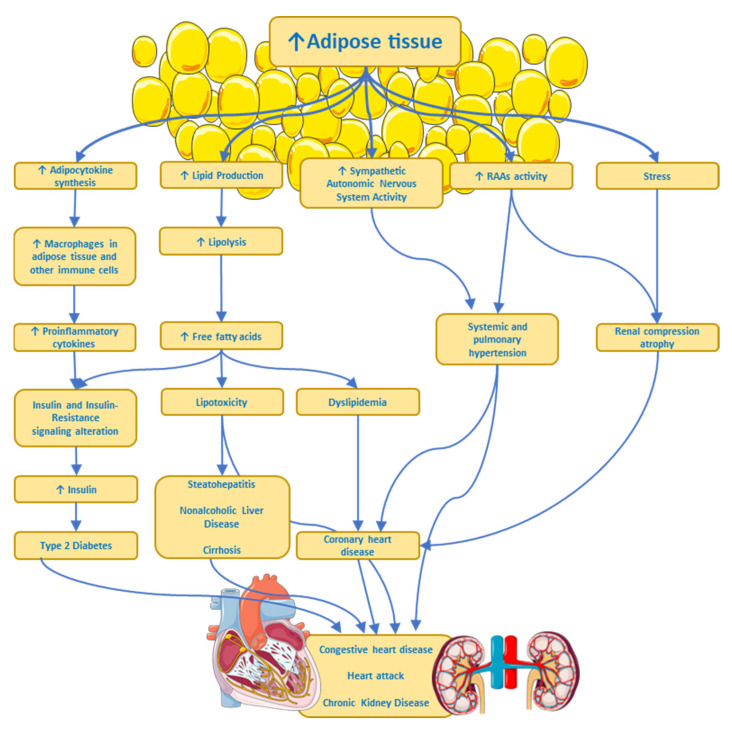
Some pathways by which excess adipose tissue produces risk factors or chronic diseases. ANS, Autonomic Nervous System; RAAs, Renin–Angiotensin–Aldosterone System. ↑ imply an increase.

**Figure 3 ijms-24-17168-f003:**
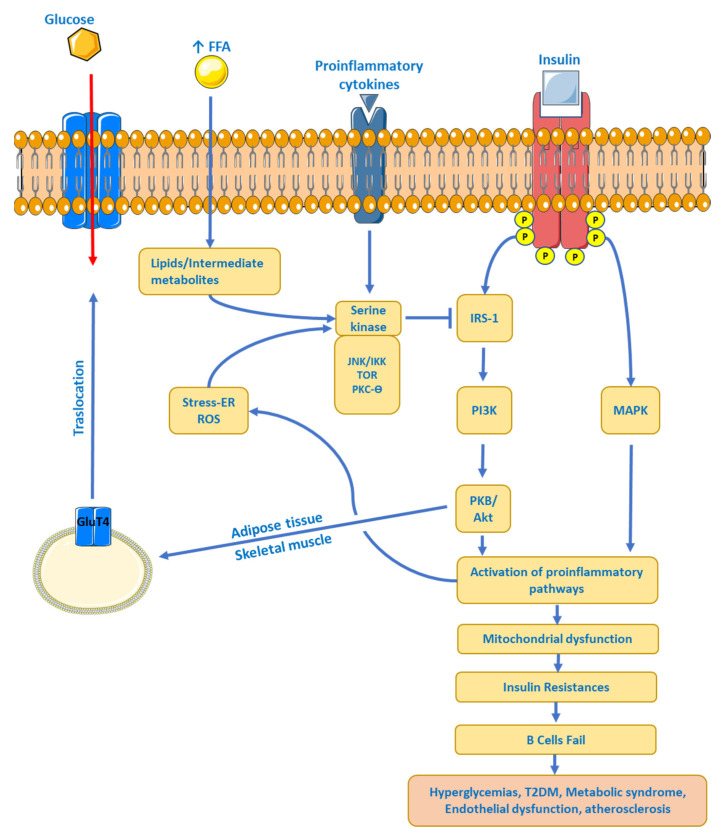
Insulin-resistance systemic level model. FFA, Free Fatty Acids; ER, Endoplasmic Reticulum; ROS, Reactive Oxygen Species; GluT4, Glucose Transporter 4; IRS-1, Insulin Receptor Substrate type 1, MAPK, Mitogen-Activated Protein Kinase; PI3K, Phosphatidyl Inositol 3 Kinase; PBK, Protein Kinase B or Akt; JNK, c-Jun N-terminal kinase; ↑ imply an increase.

**Figure 4 ijms-24-17168-f004:**
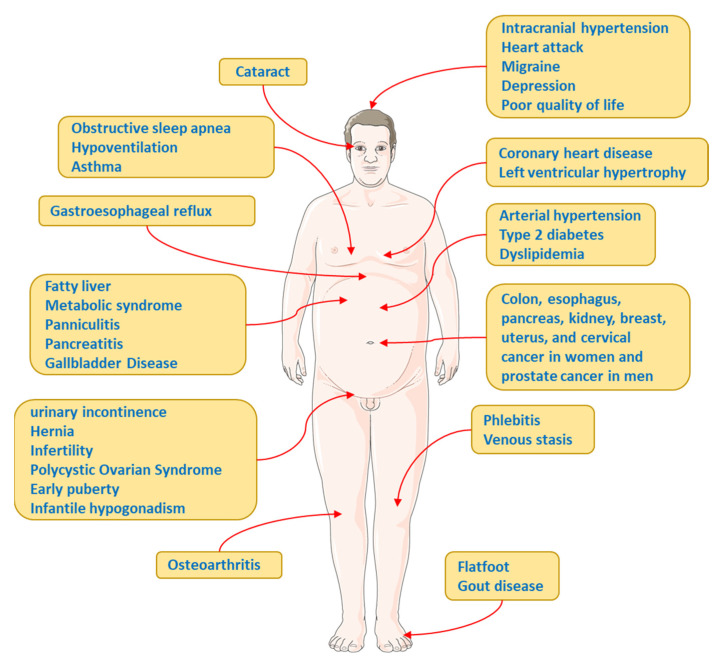
Complications and Comorbidities associated with overweight-obesity.

**Figure 5 ijms-24-17168-f005:**
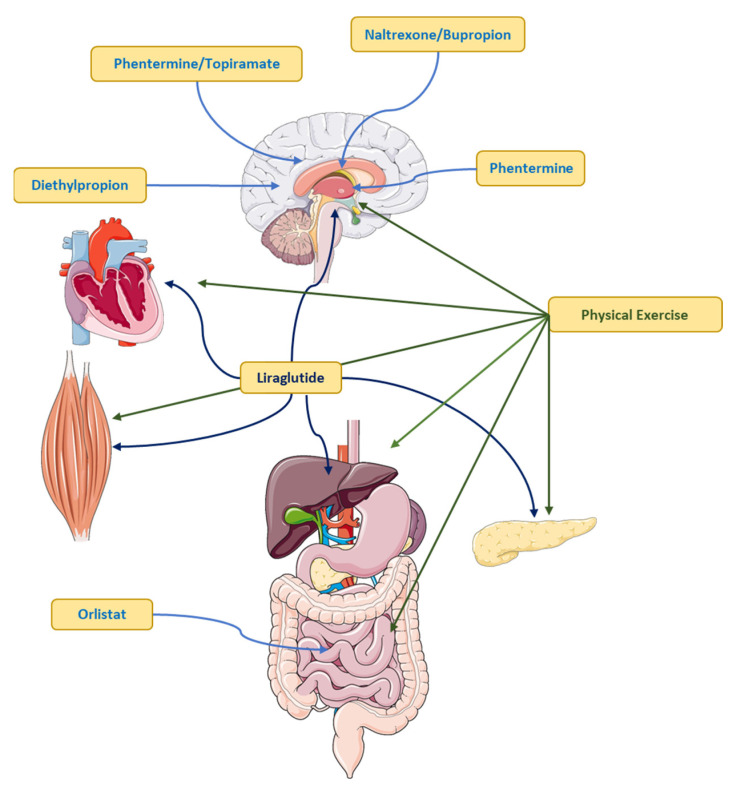
Pharmacological and non-pharmacological management of the overweight–obese patient.

**Figure 6 ijms-24-17168-f006:**
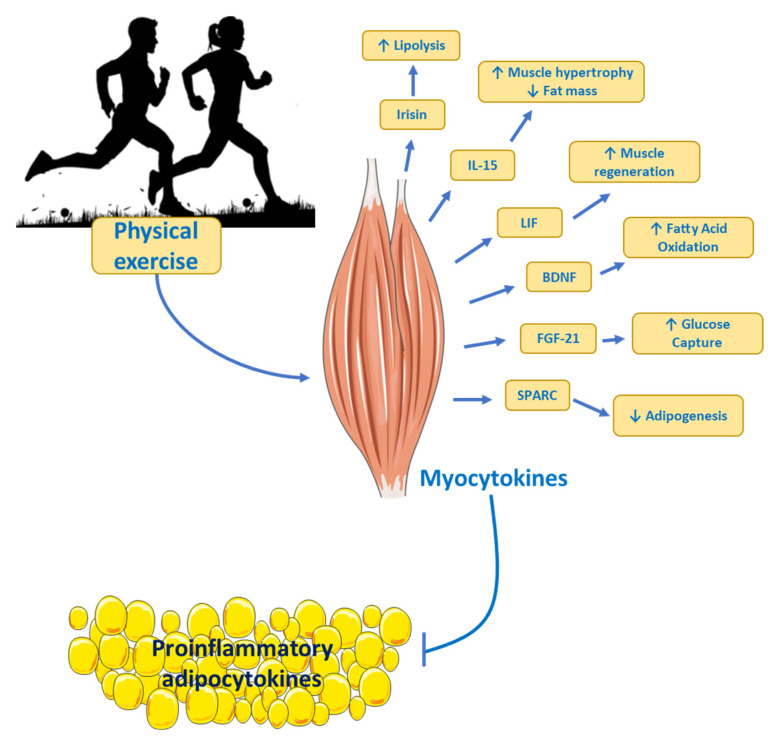
Myokines and their potential role on metabolism. Skeletal muscle produces and secretes myokines into the circulation. Adipose tissue, under conditions of metabolic diseases, secretes proinflammatory adipocytokines that promote pathological processes such as insulin resistance. However, physical exercise promotes the secretion of myocytokines which can counteract the effects of adipocytokines. IL, Interleukin; LIF, Leukemia Inhibitory Factor; BDNF, Brain-derived Neurotrophic Factor; FGF-21, Fibroblast Growth Factor 21; SPARC, Secreted Protein Rich in Acid and Cysteine; ↑ imply an increase; ↓ imply a decrease.

## Data Availability

Not applicable.

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
