# Peer review of "Adipose Tissue, Non-Communicable Diseases, and Physical Exercise: An Imperfect Triangle"

_ijms, 2023, doi:10.3390/ijms242417168_

Round 1
Reviewer 1 Report
The authors of the current review are attempting to describe the connection between adipose tissue, noncommunicable diseases, and physical activity. The abstract lacks the review's goal as well as future perspectives or ideas. It's simply a copy and paste of the major text sentences with typos. The authors wrote extensively in the main text about basic knowledge about adipose tissue and obesity that was already available in the literature. The authors' conclusions are not convincing in which way the triangle of adipose tissue, obesity, and physical activity is imperfect. The impact of their literature/information on future perspectives was not covered in the review.
Moderate editing of English language required
Author Response
Dear referees
Along with saying hello and hoping you are well. I attach a review with all the modifications suggested by you. I regret the delay in sending the document with the corrections.
Kind regards
Francisco Monsalve A., PhD.

Reviewer 2 Report
Thank you for the opportunity to read this well written, clear and concise study. It was interesting and a pleasure to read.
The subject of this article is very important, since adipose tissue is essential in many metabolic processes and a mismatch or imbalance in one of these processes can generate changes that directly affect adipose tissue. On the other hand, obesity is one of the most common diseases today that affects a greater part of the population.
Therefore, it is necessary to use new research that addresses this very current and worrying issue for public health.
For publication, authors are only asked that the references section must be adapted to the journal's standards.
Author Response

(The authors gave the same response as above.)

Reviewer 3 Report
The authors bring together a review of adipose tissue function and dysregulation. The following comments apply:
1. Imprecision in language, and spelling errors, should be eliminated; in some cases, the subject of the sentence is incorrectly placed leading to ambiguity; a thorough proof reading is recommended. Pronouns such as 'you' should be avoided, as should any colloquialisms.
2. There is a considerable degree of repetition occurring which should be removed; each sections does not need to repeat information provided in the prior section. Editing is required.
3. The sources cited are problematic in many cases. Review articles are used frequently, while primary sources are ignored: a good example here is the groundbreaking work on resistin, which certainly does not emanate from reference [17]. It is important to acknowledge the primary work in the field,a and all sources should be from peer-reviewed journals. Also - why is Jehan et al cited in the text? The authors should take particular care in reviewing this aspect of their work, in order to correctly reflect the work of key individuals in the field.
4. The idea of the lipid droplet, and its expansion, is not fully realised - for example, does the proteome differ in differing types of adipocyte? Query also the use of the term liposome - how does these differ from lipid droplets?
5. Abstract: adipose tissue is not involved in food intake - do the authors mean regulation of food intake? Sp. neuromodulatory; it is not clear whether the authors are suggesting adipokines are promoting tissue or system inflammation (or both)
6. Mechanistic details are not explored fully, for example in terms of the function of the various genes and protein contributing to obesity
7. Very low calorie diets are not considered fully as a lifestyle intervention, although they have received much attention in recent years.
Imprecisions in language occur, alongside many examples of repetitive phrases; a thorough proof reading is required to eliminate all repetition. Please also see comments to authors.
Author Response

(The authors gave the same response as above.)

Round 2
Reviewer 1 Report
The authors significantly improved the current manuscript with additional information. I recommend for the publication.